# Behavior of Inelastic and Plastic Strains in Coarse-Grained Ti$_{49.3}$Ni$_{50.7}$(at%) Alloy Deformed in B2 States

**Dorzhima Zhapova ***, **Victor Grishkov, Aleksandr Lotkov** **, Victor Timkin, Angelina Gusarenko** and **Ivan Rodionov**

Institute of Strength Physics and Materials Science of the Siberian Branch of the Russian Academy of Science, 634055 Tomsk, Russia; grish@ispms.ru (V.G.); lotkov@ispms.ru (A.L.); timk@ispms.ru (V.T.); angel.ru09@mail.ru (A.G.); rodionov231@mail.ru (I.R.)

**\*** Correspondence: dorzh@ispms.tsc.ru; Tel.: +7-382-228-69-82

**Abstract:** The regularities of the change in inelastic strain in coarse-grained samples of the Ti$_{49.3}$Ni$_{50.7}$ (at%) alloy are studied when the samples are given torsional strain in the state of the high-temperature B2 phase. During cooling and heating, the investigated samples underwent the B2–B19′ martensite transformation (MT); the temperature of the end of the reverse MT was A$_f$ = 273 K. It was found that at the temperature of isothermal cycles "loading-unloading" A$_f$ + 8 K, when the specimen is assigned a strain of 4%, the effect of superelasticity is observed. With an increase in the torsional strain, the shape memory effect is clearly manifested. It is assumed that the stabilization of the B19′ phase in unloaded samples is due to the appearance of dislocations during deformation due to high internal stresses at the interphase boundaries of the B2 phase and the martensite phase during MT. The appearance of dislocations during the loading of samples near the temperatures of forward and reverse MT can also be facilitated by the "softening" of the elastic moduli of the alloy in this temperature range. At a test temperature above A$_f$ + 26 K, the superelasticity effect dominates in the studied samples.

**Keywords:** TiNi-based alloys; superelasticity; shape memory effect; inelastic strain; plastic strain

## 1. Introduction

The significant interest in smart TiNi materials owes to the unique combination of their functional, strength, and plastic properties beneficial for engineering and medicine [1–12]. Their superelasticity (SE) and shape memory effect (SME) are provided by thermoelastic martensite transformations (MT) from a cubic B2 phase to a rhombohedral R or a monoclinic B19′ phase. When cooled and heated free of load or at low internal stress, such materials remain macroscopically invariant as their thermoelastic transformations result in a polyvariant system of self-accommodated martensite domains [4], but when exposed to external or oriented internal stresses, they display superelasticity and shape memory. Shape memory allows a material to accumulate reversible inelastic strains at T$_d$ < M$_f$ (where T$_d$ is the deformation temperature and M$_f$ is the martensite finish temperature on cooling) and to recover them at above A$_f$ (which is the austenite finish temperature on heating). Superelasticity allows a material to change its macroscopic shape or linear dimensions through direct B2 → B19′ or B2 → R → B19′ transformations under external stress (tension, torsion, bending) at T > A$_f$ and to take it back through inelastic strain recovery under subsequent isothermal unloading (because no martensite free of load can exist at this temperature). In both cases, the reversible inelastic strain can reach 6–8%. The inelastic strain, whether from SE or SME, depends largely on the crystallography limit of the recoverable strain whose value in TiNi-based alloys measures, on average, 11% [13–15].

However, reversible inelastic strains 1.5–2 times higher than the above limit have been found in TiNi-based alloys [16–23] after bending [15–18,20], torsion [20–23], and tension [19], with the total strain comprising an inelastic and a plastic component. For

example, a reversible inelastic strain of up to 18% is attainable in $Ti_{49.3}Ni_{50.7}$ (at%) after torsion at $T < M_f$ against a high (12%) plastic strain [23]. Its recovery is stepwise: via SE under unloading at $T_d$ and via SME under further heating above the finish temperature of $B19' \rightarrow B2$ MT.

In one of the cited studies [16], $Ti_{49.3}Ni_{50.7}$ (at%) specimens differing in structure were bent to a total strain of 15% and were then kept constrained at 310 K (to prevent shape recovery), cooled to 77 K with keeping at this temperature for 30 s, and unloaded with further heating to room temperature. In nanocrystalline $Ti_{49.3}Ni_{50.7}$ (at%) (grain size 30–70 nm), the total strain was recovered completely so that 7.5% fell on SE and 7.5% on SME. In $Ti_{49.3}Ni_{50.7}$ (at%) with a mixed structure (nanocrystalline, subgrain), the strain recovery via SME was 14.8% against 0.2% of plastic strain, and in its specimens with microcrystalline structure (maximum grain size 10 μm), the SME value was 10% against up to 5% of plastic strain and zero SE.

In another study [17], $Ti_{50}Ni_{50}$ (at%) specimens annealed after cold drawing were bent at 5 K above $A_f$ with further keeping at this temperature for 30 s, cooling in their constrained state to 273 K, and keeping at this temperature for 30 s. Thereafter, estimates of their strain recovery were taken under unloading (SE, including small elastic strain) and under further heating to 373 K (SME). The plastic strain is the residual one at 373 K. In $Ti_{50}Ni_{50}$ (at%) with a polygonized B2 substructure (annealing at 623 K for 1 h, subgrain size $\leq$ 200 nm), the strain recovery via SE and SME at a total strain of 18% measured 10.3% and 7.6%, respectively, against 0.1% of plastic strain; the total inelastic strain was 17.9%. In $Ti_{50}Ni_{50}$ (at%) with a polygonized B2 substructure comprising individual recrystallized grains of up to 3 μm (annealing at 723 K for 1 h), the strain recovery via SE and SME at the same total strain measured 7.2% and 8%, respectively, against 2.8% of plastic strain; the total inelastic strain reached 15.2%.

Thus, it is still unclear what conditions can bring the reversible inelastic strain in TiNi materials to above their theoretical limit. Clarifying this issue needs additional studies of TiNi alloys differing in chemical composition, structure (grain-subgrain size), and phase state. Here, we analyze the behavior of inelastic and plastic strains in coarse-grained $Ti_{49.3}Ni_{50.7}$ (at%) deformed at $T_d > A_f$.

## 2. Materials and Methods

The test material was $Ti_{49.3}Ni_{50.7}$ (at%) supplied as hot-swaged bars of diameter 30 mm (Matek-Sma Ltd., Moscow, Russia). The bars were spark cut into specimens (cross-sectional area ~1 $mm^2$, gage length ~10 mm), rinsed in ethanol, grinded with an abrasive and diamond paste, and electrolytically polished with plane stainless steel electrodes in a cold $CH_3COOH/HClO_4$ solution (75/25 vol%) at a voltage of 12–17 V for 10–15 s. For further structural analysis, they were chemically etched in a $HNO_3/HF/H_2O$ mixture (14/4/82 vol%) for 15 s.

Their structure and phase state were analyzed on a DRON-7 diffractometer in Co-$K_\alpha$ radiation (Bourevestnik JSC, Saint-Petersburg, Russia) and on an AXIOVERT-200 MAT optical microscope (Carl Zeiss AG, Oberkochen, Germany). According to the analysis, the specimens at room temperature were in the state of a high-temperature B2 phase (CsCl superstructure) with less than 5 vol% of $Ti_4Ni_2(O,N,C,H)_x$. The average grain size was $53 \pm 11$ μm. The deformation-induced surface microrelief after torsion were studied by scanning electron microscopy LEO EVO 50XVP (Carl Zeiss AG, Oberkochen, Germany). These studies were conducted using the equipment of Nanotech shared Use Center of ISPMS SB RAS.

The temperatures and sequence of martensite transformations were determined by temperature resistometry ($\rho$(T) measurements). On cooling and heating, the specimens experienced martensite transformations $B2 \leftrightarrow B19'$. The start and finish temperatures of $B2 \rightarrow B19'$ MT are $M_S = 252$ K, $M_f = 223$ K. The start and finish temperatures of $B19' \rightarrow B2$ MT are $A_S = 258$ K, $A_f = 273$ K.

The inelastic and plastic strains in the material were studied on an inverted torsion pendulum with an operating temperature of 573–120 K. The τ–γ dependences in isothermal loading–unloading cycles, and the inelastic strain recovery on further heating to 500 K (227 K above $A_f$) of unloaded specimens were obtained. In each cycle, the total strain $\gamma_t$ was successively increased up to fracture. The temperature of loading–unloading cycles was 281, 299, 309, 315, and 339 K, i.e., $T_d$ was above $A_f$ and specimens were in the B2 state.

All components of the total strain $\gamma_t$ were determined. The total strain is the sum of inelastic and plastic strains: $\gamma_t = \gamma_{SID} + \gamma_{pl}$. The total inelastic strain is $\gamma_{SID} = \gamma_{SE} + \gamma_{SME}$, where the summands stand for inelastic strains recovered via superelasticity, $\gamma_{SE}$, under isothermal unloading (including a small Hook strain of ~1.5%) and via shape memory, $\gamma_{SME}$, under further heating to complete shape recovery (via B19′ → B2 transformation). In more detail, the total inelastic strain has the form:

$$\gamma_{SID} = (\gamma_t - \gamma_r) + (\gamma_r - \gamma_{pl}) = \gamma_{SE} + \gamma_{SME} \tag{1}$$

where $\gamma_r$ is the residual strain after isothermal unloading (295 K), and $\gamma_t$, $\gamma_r$, and $\gamma_{pl}$ are equal to arctg$S_t$, arctg$S_r$, and arctg$S_{pl}$, with $S_t = (r\varphi_t)/l$; $S_r = (r\varphi_r)/l$; and $S_{pl} = (r\varphi_{pl})/l$; having r and l for the specimen cross-section radius and gage length and $\varphi_t$, $\varphi_r$, and $\varphi_{pl}$ for the torsion angles in radians after loading, unloading, and heating to 500 K, respectively. The plastic strain $\gamma_{pl}$ is equal to the residual strain at 500 K. The $\gamma_t$, $\gamma_r$, $\gamma_{pl}$, $\gamma_{SE}$ and $\gamma_{SME}$ strains are presented in Figure 1. The measurement error for the quantities depended on the total strain, and at $\gamma_t = 36.6\%$, it was $\Delta\gamma_t = 0.3\%$, $\Delta\gamma_{SE} = 0.4\%$, $\Delta\gamma_{SME} = 0.4\%$, and $\Delta\gamma_{pl} = 0.2\%$.

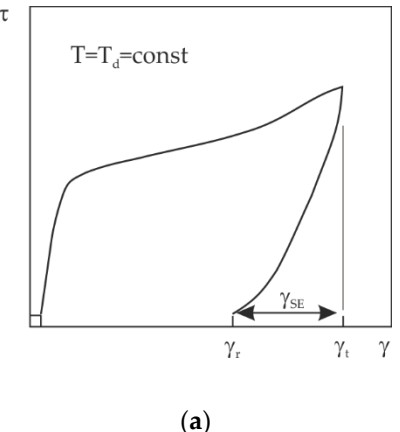 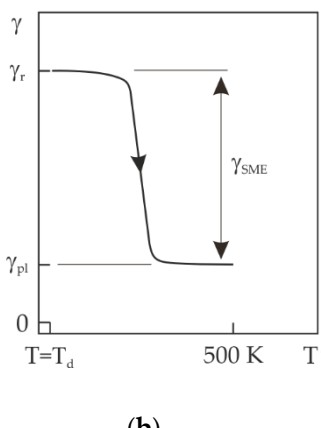

(**a**)  (**b**)

**Figure 1.** The experimental strains determined in isothermal τ–γ cycle (**a**) and on subsequent heating of unloaded specimen (**b**).

## 3. Results

Figure 2 shows the τ–γ dependences of Ti$_{49.3}$Ni$_{50.7}$ (at%) in loading-unloading cycles at 281, 299, 309, and 339 K. As can be seen, the dependences at 281 and 299 K reveal four deformation stages. Stage I is quasi-elastic and its strain increases linearly as the applied stress is increased. At τ = $\tau_m$ (martensite shear stress), stage I passes into stage II, at which the strain increment is large while the external stress build-up is comparatively small. Stage II represents a so-called pseudo-yield plateau which ends, in our case, at about $\gamma_t = 8\%$. Obviously, at 281 and 299 K (Figure 2, curves 1, 2), the pseudo-yield plateau results from the generation of B19′ martensite under loading [4,13]. Stage II is followed by stage III, at which $\gamma_t$ increases almost linearly from 8% to 18–20% as τ is increased from ~450 to 800 MPa. Stage III represents strain hardening, which passes into active plastic flow at stage IV. From Figure 2, it is seen that when deformed at 309 K (curve 3) and 339 K (curve 4), the material shows no pseudo-yield plateau and its quasi-elastic stage passes directly into parabolic flow.

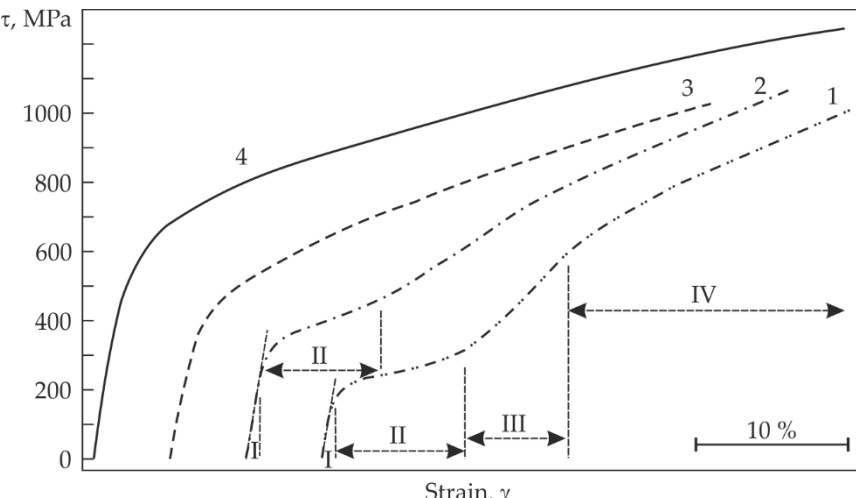

**Figure 2.** Stress–strain dependence of Ti$_{49.3}$Ni$_{50.7}$ (at%) specimens deformed at 281 K (1), 299 K (2), 309 K (3), and 339 K (4). The start strain for each $\tau$–$\gamma$ dependence corresponds $\gamma = 0$.

Figures 3 and 4 show the $\tau - \gamma$ dependences of Ti$_{49.3}$Ni$_{50.7}$·(at%) in loading–unloading cycles with successively increasing $\gamma_t$ and its inelastic strain recovery on further heating at different test temperatures. From Figure 3a, it is seen that as the number of loading–unloading cycles with $T_d = 281$ K is increased, the martensite shear stress $\tau_m$ grows and the extent of the pseudo-yield plateau first increases, then decreases (after the fifth cycle). It should be noted that the martensite shear stress $\tau_m$ increases from 180 MPa in the first loading–unloading cycle to 380 MPa in the seventh one. After the first loading–unloading cycle with $\gamma_t = 4.1\%$, the strain recovery via superelasticity (under unloading) was 3.9%, and via shape memory (at heating), it was 0.2%. Although $T_d$ was higher than $A_f$ by about 8 K, the SE value in the next isothermal $\tau - \gamma$ cycles was low compared to $\gamma_t$. Even after the second cycle with $\gamma_t = 9.9\%$, the SE value was $\gamma_{SE} = 3.6\%$ against $\gamma_{SME} = 6.3\%$ (Figure 3a), and no plastic component was detected within the measurement error. After the third cycle with $\gamma_t = 14.4\%$, the SE and SME values were $\gamma_{SE} = 3.0\%$ and $\gamma_{SME} = 11.0\%$ against $\gamma_{pl} = 0.4\%$ (Figure 3b). As the number of loading–unloading–heating cycles was further increased, $\gamma_{SE}$ increased monotonically. The $\gamma_{SME}$ value first increased and then decreased, and $\gamma_{pl}$ became larger (Figure 3b).

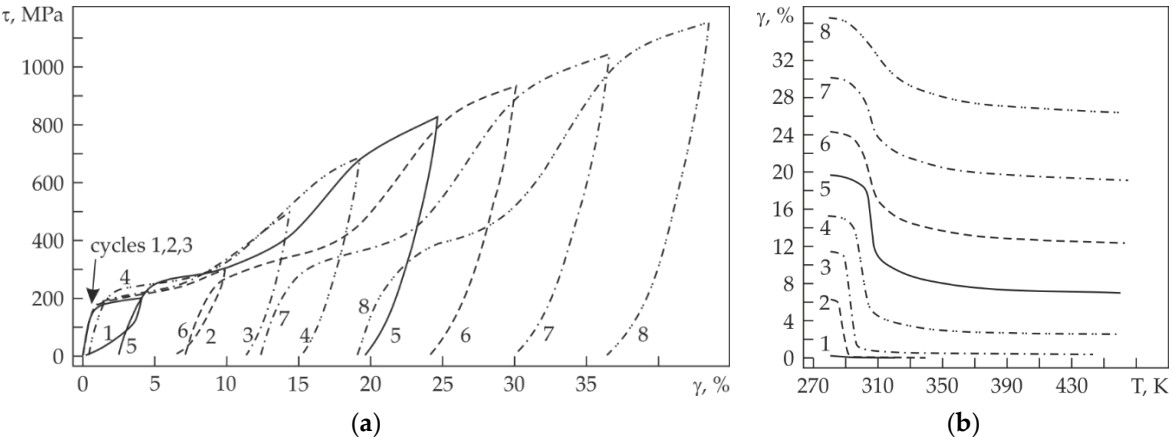

**Figure 3.** Accumulation and recovery of inelastic strain in loading–unloading cycles at 281 K (**a**) and its recovery on further heating (**b**).

From Figure 4a it is seen that at $T_d = 299$ K (26 K above $A_f$), the first two loading–unloading cycles with increasing $\gamma_t$ result in flag-shaped $\tau - \gamma$ dependences characteristic of superelasticity so that the total strain is fully recovered after unloading. It should

be noted that all loading–unloading cycles at this temperature are dominated by the SE effect. After the first cycle with $\gamma_t$ = 3.7%, $\gamma_{SE}$ = 3.6%, and after the second cycle with $\gamma_t$ = 7.3%, its value is $\gamma_{SE}$ = 6.9%, i.e., almost the whole strain is recovered under unloading. After the third cycle, $\gamma_{SE}$ is equal to about 10.6%, but some strain remains unrecovered under unloading. As the number of loading–unloading cycles as well as the total strain is further increased, $\gamma_{SE}$ varies little. It should be noted that at $T_d$ = 299 K, compared to $T_d$ = 281 K, the SE effect is much more pronounced and the SME value $\gamma_{SME}$ on heating is small (Figure 4b). It should also be noted that after the third loading–unloading cycle with up to $\gamma_t$ = 12.3%, the strain is mostly inelastic and almost completely recovered via SE and SME under unloading and heating, respectively. However, in the next cycles, the plastic strain component steeply increases. Noteworthy also is that the pseudo-yield plateau at $T_d$ = 299 K is poorly detectable in all loading–unloading cycles (Figure 4a).

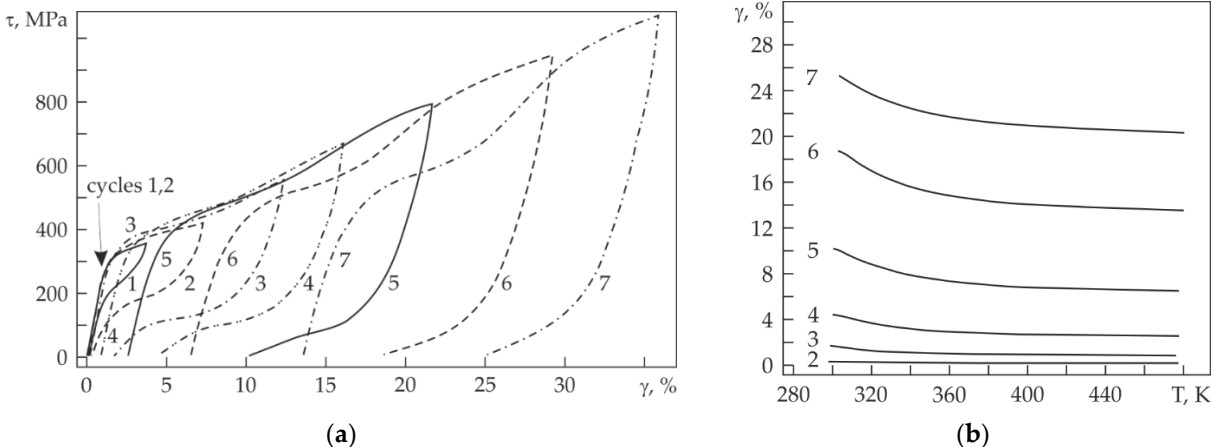

**Figure 4.** Accumulation and recovery of inelastic strain in loading–unloading cycles at 299 K (**a**) and its recovery on further heating (**b**).

Figure 5 shows the $\gamma_t$ dependences of $\gamma_{SID}$, $\gamma_{SE}$, and $\gamma_{SME}$ at different temperatures of isothermal $\tau-\gamma$ cycles. It is seen that at 281 K, $\gamma_{SME}$ reaches 12.7% (Figure 5a). At the same time, the temperature interval of shape recovery is narrow on heating, spanning from 285 to 310 K (Figure 3b). Note that at all temperatures of isothermal loading–unloading cycles, except for $T_d$ = 281 K, the $\gamma_t$ dependences of $\gamma_{SID}$, $\gamma_{SE}$, and $\gamma_{SME}$ are qualitatively similar: as $\gamma_t$ is increased, these quantities first reach a maximum and then decrease. The $\gamma_t$ strains at which $\gamma_{SID}$, $\gamma_{SE}$, and $\gamma_{SME}$ reach their maximum values differ but fall on the stage of active plastic strain accumulation (Figure 5).

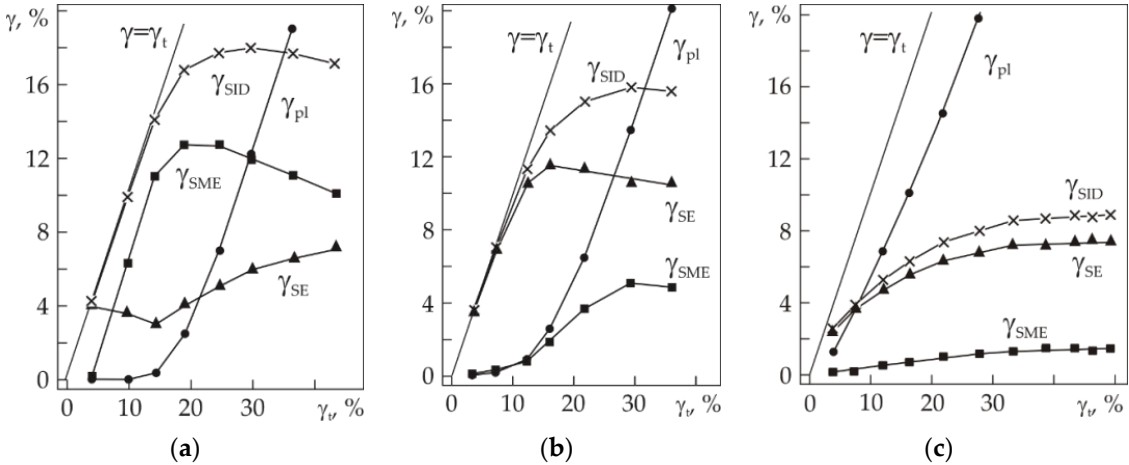

**Figure 5.** Dependences of $\gamma_{SE}$ (▲), $\gamma_{SME}$ (■), $\gamma_{SID}$ (×), and $\gamma_{pl}$ (●) on $\gamma_t$ after isothermal loading-unloading at 281 K (**a**), 299 K (**b**), 339 K (**c**) and subsequent heating of unloaded specimens.

## 4. Discussion

Certainly, at first glance, it is rather strange that the total strain measuring about a mere 4.1% at 281 K (8 K above $A_f$) results in an SME of 0.2% (Figure 3b). With such a small total strain at 281 K, one would expect its complete recovery via SE after unloading [24]. Moreover, after the second loading–unloading cycle with $\gamma_t$ = 9.9%, the SE value decreases to 3.6% against 3.9% after the first cycle, whereas the SME value increases from 0.2% in the first cycle to 6.3% in the second one (Figure 3b). Although no plastic strain is found in the second loading–unloading cycle, the rise of SME after the first and second cycles at 8 K above $A_f$ is most likely due to the generation of dislocations at the interfaces of B2 and B19$'$ phases under internal stress, which results from their lattice misfit [25]. It has been known that as the temperature of TiNi-based alloys is decreased and brought to about $M_S$, their elastic constants C$'$ and $C_{44}$ decrease greatly [26–28], and the decrease in the elastic constants should decrease the martensite shear stress in this temperature range. It should be noted that at $T_d > A_f$ in the temperature range of decreased elastic constants, the recovery of inelastic strains via SE at comparatively low external stresses can be incomplete due to the rise of small plastic strains lying within 0.3%. The stabilization of martensite B19$'$ caused by the development of plastic microdeformations as a result of the appearance of dislocations at small, specified strains in binary alloys with ~50 and 50.8 at% Ni was observed in [29–31]. The generation of lattice defects and, primarily, dislocations during plastic deformation and martensite transformations impedes the recovery of inelastic strains via SE and can even block this process. On heating after unloading, the recovery is contributed to by SME. Thus, it is reasonable to suggest that the rise of SME in the test material after the first and second loading–unloading cycles at 281 K, which is 8 K higher than $A_f$, is due to dislocation generation at comparatively low stresses. When the temperature is increased to 299 K, the strain recovery after the second cycle with $\gamma_t$ = 7.2% measures 6.9% via SE and only 0.1% via SME against 0.2% of plastic strain (Figures 4 and 5b). In addition, the analysis of experimental data shows that increasing $T_d$ from 309 to 339 K leads to a decrease in $\gamma_{SE}$ from 6% to 4% in isothermal cycles $\tau$–$\gamma$ with $\gamma_t \approx 7\%$. At the same time, the residual deformation increases from 1% to 3% after isothermal unloading of samples. Hence, the temperature interval of manifestation of superelasticity (4–7% and practically without residual deformation) is very narrow and localized from 281 K ($A_f$ + 8 K) to 299 K ($A_f$ + 26 K).

The fact that in loading–unloading cycles at $T_d \geq 309$ K, the SE and the SME value decreases and the plastic component steeply increases, is explained by the deformation temperature approaching $M_d$, at which no B2 $\rightarrow$ B19$'$ transformation is possible in the material under applied stress. The higher the temperature of isothermal loading–unloading cycles, the lower the total inelastic strain $\gamma_{SID}$ and the higher the plastic component $\gamma_{pl}$ (Figure 5). In this work, we did not investigate the nature of the development of reversible and plastic strains during torsion of the samples (nonhomogeneous or homogeneous [32]). At the same time, it was shown in [31] that the plastic strain develops by dislocation sliding and twinning in an alloy of close composition (50.8 at% Ni) with an average grain size of 500 nm.

Plastic deformation accumulated during torsion causes the appearance of deformation microrelief on the original polished surface of the samples. Figure 6 shows the specimen surface microrelief after torsion at 299 K. Dependences of inelastic deformations $\gamma_{SE}$, $\gamma_{SME}$, $\gamma_{SID}$ and plastic deformation $\gamma_{pl}$ on a total $\gamma_{pl}$ were presented in Figure 6. The surface microrelief image was obtained after the final cycle $\tau$–$\gamma$ with $\gamma_t$ = 38%, heating the unloaded sample to 500 K and subsequent cooling to 299 K. The sample after cooling had a B2 phase structure. Plastic deformation $\gamma_{pl} \approx 20\%$, Figure 5c. Figure 6 clearly shows large deformation fragments with an intense microband structure within them. The average size of the large fragments, estimated from three similar images, is 46 $\pm$ 5 $\mu$m. This value correlates well with the average size of the original grains (53 $\pm$ 11 $\mu$m). Similar deformation fragments with sizes equal to those of the original grains arise as a result of mutual reversals and displacements of neighboring grains relative to each other during tensile

plastic deformation of various materials not undergoing martensitic transformations: for example, aluminum-based alloy (Al 1100-0 [33], Al-Mg [34], AA-6022 [35]), AISI 1010 [36] steel, and others. At the same time, the formation of a fine microband structure within the fragments is due to the development of intra-grain plastic deformation processes by the mechanisms of dislocation sliding [29,31].

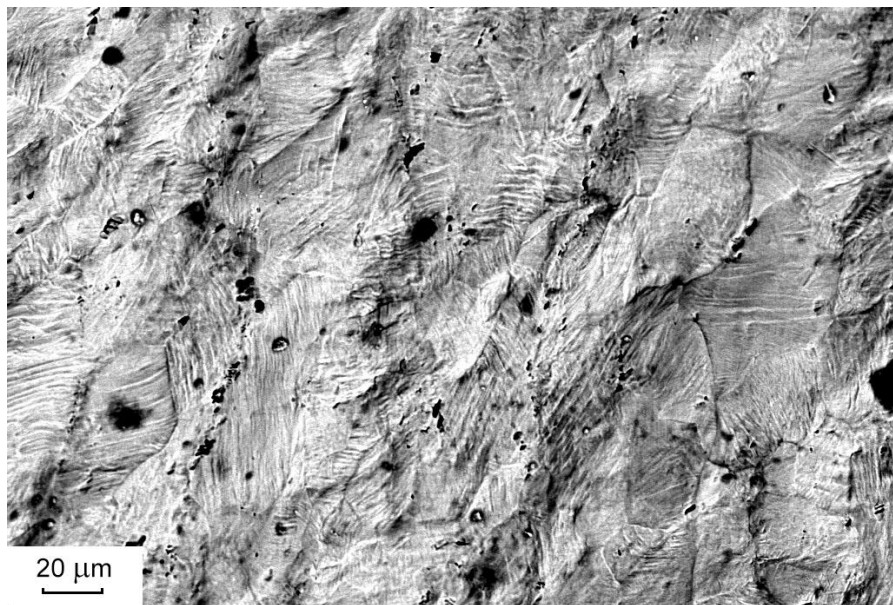

**Figure 6.** Deformation-induced surface microrelief after torsion at 299 K of coarse-grained specimen of $Ti_{49.3}Ni_{50.7}$ (at%) alloy (details in text). Scanning electronic microscopy.

The maximum total reversible inelastic strain in torsion is $\gamma_{SID}$ = 18% and is attained at 281 K (8 K higher than $A_f$) with $\gamma_{pl}\cdot$ = 12% against $\gamma_t \approx$ 30%. Interestingly, the same result is reported for $Ti_{49.3}Ni_{50.7}$ (at%) after torsion at T < $M_f$ [23]: $\gamma_{SID}$ = 18% with $\gamma_{pl}$ = 12%. We think that this coincidence is not by chance but results from a certain common mechanism.

In conclusion, it should be noted that a direct comparison of inelastic and plastic strains obtained under torsion with similar strains obtained under compression, tension, or bending, and with the crystallography limit of martensitic strains at $B19' \leftrightarrow B2$ MT, which is determined by compression-tension of the initial phase crystal lattice, is impossible. This is due to the difference in the deformation modes and the determination of the corresponding strains. In the first approximation, for a correct comparison of the total inelastic strain and plastic strain obtained in torsion with the results of studies of similar strains obtained in tension and bending and the crystallography limit of martensitic strain at $B2 \leftrightarrow B19'$ MT, equal to ~10% for the polycrystalline $Ti_{49.3}Ni_{50.7}$ (at%) alloy [14], it was the concept of equivalent true strain at different loading modes, used earlier in [20]. True tensile strain $e_1 = \ln(1+\varepsilon)$, where $\varepsilon$ is the relative elongation. True torsional strain $e_2 = S/\sqrt{3}$, where $S = tg\gamma$ is the accumulated shear strain. Tensile strains $\varepsilon_t$, $\varepsilon_r$, $\varepsilon_{pl}$, corresponding to torsional strains $\gamma_t$, $\gamma_r$, $\gamma_{pl}$, can be obtained from the relation $e_1 = e_2$. Using these values of $\varepsilon_t$, $\varepsilon_r$, and $\varepsilon_{pl}$, the inelastic strains recovered by the realization of SE, $\varepsilon_{SE}$, and SME, $\varepsilon_{SME}$, and the total inelastic strain $\varepsilon_{SID}$ can be determined. The tensile strains corresponding to torsion strains after loading at 281K ($\gamma_{SID}\cdot$ = 18%, $\gamma_{pl}$ = 12%, $\gamma_t\cdot$ = 30%) are: $\varepsilon_t$ = 19.6%, $\varepsilon_{SID}\cdot$ = 12.4%, $\varepsilon_{pl}\cdot$ = 7.2%. To take into account that both $\gamma_{SID}$ and $\varepsilon_{SID}$ include Hooke's elastic strain, it can be concluded that the maximum reversible inelastic strain $\gamma_{SID}$ during torsion approximately corresponds to the known crystallography limit of martensitic strain at $B2 \leftrightarrow B19'$ MT in our coarse-grained alloy samples with 50.7 at% Ni. This result corresponds to the data of [16,31]. When bending samples of alloy $Ti_{49.3}Ni_{50.7}$ (at%) with microcrystalline structure (grain size $\leq$ 10 µm), the maximum inelastic strain equal also to 10% was obtained [16]. It was shown in [31] that the maximum total recoverable inelastic

strain $\varepsilon_{SID}$ in σ-ε cycles during tension and the subsequent heating of unloaded binary alloy samples with 50.9 at% Ni and an average grain size of 500 nm is 13%, which is close to the maximum total inelastic strain in our coarse-grained alloy with 50.7 at% Ni. Thus, abnormally high inelastic strains (~18% [23] and ~15% [16] obtained under the bending of ultrafine-grained TiNi-based binary alloys) are not observed in the coarse-grained samples of binary alloy with 50.7 at% Ni.

## 5. Conclusions

It was found that the maximum value of the total inelastic torsional strain ($\gamma_{SID}$) of coarse-grained samples of $Ti_{49.3}Ni_{50.7}$ (at%) alloy, which was achieved in this work, is 18% and is observed with developed plastic deformation of about 12% (a total specified deformation $\gamma_t \approx 30\%$). It was shown that the maximum value of total inelastic strain $\gamma_{SID}$· = 18% obtained during torsion corresponds to the crystallographic limit of martensitic strain at B2 $\leftrightarrow$ B19′ MT.

It was shown that very narrow temperature range of a bright manifestation of the superelasticity effect (4–7% and practically without the residual strain) in coarse-grained specimens of $Ti_{49.3}Ni_{50.7}$ (at%) alloy is from 281 K ($A_f$ + 8 K) to 299 K ($A_f$ + 26 K).

It is shown that in the temperature range from $A_f$ + 8 K to $A_f$ + 26 K, the maximum inelastic strain that can be obtained in coarse-grained samples of alloy $Ti_{49.3}Ni_{50.7}$ (at%) during torsion and can be returned completely as a superelasticity effect is 4–7%. When the external stress is increased in order to achieve a larger value of the superelasticity effect, the yield strength of the samples is exceeded. This leads to plastic strain of samples and hinders the return of inelastic strain in the form of superelasticity effect. This is the reason for the appearance and increase in SME when the $\gamma_t$ total strain increases under torsion of samples at temperatures above $A_f$.

**Author Contributions:** Conceptualization, D.Z., V.G., and A.L.; investigations, D.Z., I.R., A.G., and V.T.; writing–original draft preparation, D.Z.; writing–review and editing, D.Z. and A.L.; project administration, D.Z. and V.G.; funding acquisition, D.Z. All authors have read and agreed to the published version of the manuscript.

**Funding:** This research was funded by Government research assignment for ISPMS SB RAS (project FWRW-2021-0004) and a grant of the President of the Russian Federation No. MK-1057.2020.8.

**Data Availability Statement:** Not applicable.

**Conflicts of Interest:** The authors declare no conflict of interest.

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
