# Peer review of "Behavior of Inelastic and Plastic Strains in Coarse-Grained Ti49.3Ni50.7(at%) Alloy Deformed in B2 States"

_metals, doi:10.3390/met11050741_

Round 1
Reviewer 1 Report
The article describes behavior of reversible (superelastic and shape memory) strains and plastic strain in coarse-grained NiTi deformed in torsion. The article brings interesting experimental results which well correspond with current works dedicated to stability and plastic deformation mechanisms in NiTi shape memory alloys. Prior to possible publication I recommend to improve the paper in the following points:
- Figure 2: A reason for different initial deformations is not clear. I would also add some labels of x-axis.
- I appreciate the experimental results presented, however, the discussion of the results does not reflect the current state of knowledge of mechanisms of plastic deformation in NiTi and I miss references to several recent works in this topic:
- I miss reference to the classical paper about martensite stabilization in NiTi deformed in torsion by [Liu2000].
- Recently, several similar works where reported in literature describing recovering of SE and SME strains induced in tension – see e.g. [Chen2019]. These works are complemented by more detailed analysis of deformation mechanisms and clearly show that deformation twinning plays a prominent role in plastic deformation of B19’ martensite. A theoretical explanation of this twinning can be found e.g. in [Gao2017]. I strongly recommend to reflect these new findings in your discussion.
- I recommend to place the note about equivalence of strains at different loading modes already at the beginning of the article. I agree that direct comparison of torsion with different loading modes is problematic, however, the concept of equivalent strain is usually applied and allows to estimate equivalent tensile strain. Please, explain also how you recalculated your values of strains (30%, 18% and 12%) resulting in tensile equivalent (19.6%, 12.4% and 7.2%) – I supposed that equivalent strain in tension is equal to the strain in torsion divided by 3^0.5, however, it does not fit you results.
[Liu2000] Liu, Y., Favier, D., Stabilisation of martensite due to shear deformation via variant reorientation in polycrystalline NiTi (2000) Acta Materialia, 48 (13), pp. 3489-3499.
[Chen2019] Chen, Y., Molnárová, O., Tyc, O., KadeÅ™ávek, L., Heller, L., Šittner, P., Recoverability of large strains and deformation twinning in martensite during tensile deformation of NiTi shape memory alloy polycrystals (2019) Acta Materialia, 180, pp. 243-259.
[Gao2017] Gao Y, Casalena L, Bowers M, Noebe RD, Mills MJ, Wang Y (2017) An origin of functional fatigue of shape memory alloys. Acta Mater. 126:389–400.
Author Response
Dear Reviewer!
The authors thank you for your attention to our work. We agree with your comments and have tried to make appropriate changes in the manuscript.
Sincerely yours
Dorzhima Zhapova.

Reviewer 2 Report
Present study provides the experimental result of plastic torsion in a nitinol SMA. In principle, the stress-induced martensitic transformation and also plasticity of dislocation glide begin at surface and develops by moving the elastic and inelastic boundaries. The inhomogeneity of stress and strain should be considered in the discussion of SE, SME, and plastic residual strains. As for the torsion of SMA, there are some previous studies, [1]~[3] for example. These should be referred to.
[1] Phase transformation in superelastic NiTi polycrystalline micro-tubes under tension and torsion––from localization to homogeneous deformation, Q-P Sun and Zhi-Q Li , International Journal of Solids and Structures 39(2002), 3797-3809
[2] Cyclic torsional loading of pseudoelastic NiTi shape memory alloys: Damping and fatigue failure
W. Predki M. Klönne A. Knopik, Materials Science and Engineering: A 417(2006) 182-189
https://doi.org/10.1016/j.msea.2005.10.037
[3] Torsional Behavior of NiTi Wires and Tubes: Modeling and Experimentation, C. Chapman, A. Eshghinejad, M. Elahinia
J Intelligent Material Systems Structures, 22(2011)
https://doi.org/10.1177/1045389X11411224
Author Response
Dear Reviewer!
The authors are grateful for your attention to our work. We have made appropriate additions to the text of the manuscript.
We are grateful for your recommended works, which give us grounds for further research in this direction.
Sincerely yours
Dorzhima Zhapova.

Round 2
Reviewer 2 Report
Comments #2
Our interest concerns the magnitude of inelastic torsion strain, the stage II observed in this study. The principal tensile strain of the transformation lattice strain of NiTi martensite is 11 % [1]. When the stress-induced transformation under tension is accompanied by slip, the inelastic plateau stage was equal to the principal strain [2]. The inelasticity strain in shear would be twice as large as the principal lattice strain. As compared with 15 % strain in Ref [15], I wonder if the shear strain of 6~10 % in Fig.4 appears to be too small. Some reasoning would be needed.
[1]Sehitoglu et al. Metal Mater Trans A 34A(2003) 5-13. [2]Kato, Sasaki, INTPLA 50(2013) 37-48.
L.199-202 / I understand and agree with the sentence.
L.221-225 / The scientific importance of this paper is the torsion. Tensile testing is easy to perform, and numerous results have been published.
L.259-265./ The term “shear intensity” is not familiar to me.
L.272-276/ The discussion should be given in Introduction.  /EOF
Author Response
We are grateful to you for your attention to our work and the comments you have made.
